# Prefrontal Cortex Activation during Memory Training by Virtual Drum Beating: A Randomized Controlled Trial

**DOI:** 10.3390/healthcare11182559

**Published:** 2023-09-15

**Authors:** Yeon-Gyo Nam, Bum-Sun Kwon

**Affiliations:** 1Department of Physical Therapy, Sun Moon University, Asan 31460, Republic of Korea; nyg35830@sunmoon.ac.kr; 2Department of Rehabilitation Medicine, School of Medicine, Dongguk University, Goyang 10326, Republic of Korea

**Keywords:** cognitive health, exergames, VR/AR, clinical training

## Abstract

The use of virtual reality (VR) content in neurological disorders with cognitive impairment is increasing. We have developed a device that incorporates virtual drum beating content, designed for digit memorization training. This study aimed to investigate the effects of realistic cognitive training on brain activity using functional near-infrared spectroscopy (fNIRS). Thirty healthy individuals were recruited and randomly assigned into two groups: conventional cognitive exercise (CCE) and a realistic cognitive exergame (RCE). Subjects in the CCE group underwent memory training by memorizing numbers displayed on a computer screen and then writing them on paper. The main outcome measure was the oxyhemoglobin level in the dorsolateral prefrontal cortex (DLPFC). As a result, the average number of digits was 7.86 ± 0.63 for the CCE and 7.6 ± 0.82 for the RCE. The mean difference in ΔHbO was 1.417 ± 0.616 μm (*p* = 0.029) in channel 2, located in the right DLPFC. Channel 7 and channel 10, which measured activations in the hypothesized medial orbitofrontal cortex (OFC), also showed a significant mean difference of ΔHbO. DLPFC and OFC presented higher activation in the RCE group (*p* < 0.05), attributable to the simultaneous memory training and virtual drum beating, which provided various sensory inputs (visual, auditory, and vibration). Although DLPFC involvement in cognitive processes remains controversial, our findings suggest that realistic memory training using drumming content can lead to safer activation of the DLPFC compared to conventional cognitive training.

## 1. Introduction

Cognitive changes are a normal part of the aging process, affecting individuals differently. It is estimated that approximately 15% to 20% of individuals aged 65 and older experience mild cognitive impairment (MCI) [1]. Given the desire to prevent or slow down further cognitive decline in individuals with MCI, there is significant interest in potential treatments [2]. One such treatment approach is cognitive training (CT), which involves structured and repetitive mental exercises aimed at improving specific cognitive processes [3]. CT is a non-pharmacological treatment method that focuses on the guided practice of tasks that target specific cognitive functions, such as memory and attention [4]. A previous review study has shown that CT can lead to significant improvements in memory, executive function, processing speed, attention, fluid intelligence, and the subjective evaluation of cognitive functioning among older adults with MCI [5]. Moreover, several interventions are focused on cognitive training, and many studies have been conducted to investigate their effects [4]. However, it remains unclear whether recently developed CT can effectively assist older adults with MCI in maintaining or improving their cognitive abilities, overall well-being, and general functioning [4].

Memory training content with virtual reality (VR) is one of the most advanced Information Communications Technologies (ICT), offering an immersive, intuitive, motivating, interactive, and multisensory feedback context [6]. VR refers to complete three-dimensional virtual representations of real-world scenarios or objects [7]. The combination of VR with neuroimaging is increasingly being explored in the clinical field due to its potential to enhance neuropsychological treatments [8]. Previous studies have shown the positive effects of VR-based gaming interventions on attention, visual and verbal memory, and executive function in older adults with MCI [9]. A fully immersive VR gaming system can incorporate a wide field of view in high-resolution graphics on a head-mounted display (HMD) with auditory feedback [10]. Moreover, VR applications delivered in an HMD are widely used for rehabilitation, assessment, and even the prediction of cognitive impairments in older adults [11]. However, some reviews have addressed potential adverse effects associated with HMD usage, often referred to as “VR sickness”. This term encompasses a range of symptoms and their severity that are commonly reported by individuals using HMDs [12]. Such symptoms can include feelings of nausea, dizziness, and blurred vision, which have been documented in relation to HMD usage [13].

Recently, realistic cognitive training content involving drumming was developed especially for memory training in older adults. To prevent the risk of VR sickness, this training system deliberately omitted the use of HMD. The focus was to minimize the potential for VR sickness and create a user-friendly experience for older adults and patients with MCI. The content was developed by prioritizing the immersive sensation of playing a musical instrument, replicating the experience of playing a physical drum. To execute the training activities, participants used a handheld controller, which allowed them to interact with the content. The system incorporated tactile feedback through vibrations and auditory cues, simulating the authentic sensory experience of playing a physical drum within the virtual drumming environment.

fMRI is a well-known and the most used tool in neuroimaging studies, providing the most precise test in mapping functional brain activation patterns across the entire brain [14]. During fMRI scans, subjects are generally instructed to stay as motionless as possible, as even minimal body movement can alter or compromise the quality of the data obtained [15]. fMRI is inappropriate to use when subjects are engaged in limb movement, as it cannot effectively mitigate motion-related artifacts [16]. On the other hand, fNIRS is a non-invasive optical neuroimaging method renowned for its ability to provide powerful data even in the presence of movement artifacts. This makes it particularly valuable in studies involving the free movement of limbs [17]. For instance, a 15-channel fNIRS probe configuration can effectively map each channel to different parts of the Brodmann areas when placed on the prefrontal cortex. Among the various neuroanatomical regions of the brain, the dorsolateral prefrontal cortex, the ventrolateral prefrontal cortex, and the frontopolar cortex have been primary subjects of investigation [18]. Current evidence highlights the important role of the frontal lobe, one of the brain’s most significant neuroanatomical structures, particularly in attention processes during cognitive tasks [19].

No prior research has explored the potential effects of instrument playing with fNIRS on brain activity, particularly following hemoglobin pathways. In the present study, we used fNIRS to assess changes in brain activity during virtual drumming exercises. Although the virtual drumming content and equipment were initially designed for older adults’ cognitive training, the present study was conducted on healthy people without MCI as the research was a preliminary study. To our knowledge, this is the first study to investigate differences in brain activity during memory training between conventional methods and a realistic cognitive exergame without causing VR sickness.

## 2. Materials and Methods

### 2.1. Study Protocol and Study Design

This study was a randomized controlled trial, single-blinded (assessor), investigating differences in brain activity between conventional cognitive exercise (CCE) and a realistic cognitive exergame (RCE). The research protocol was approved by the Institutional Review Board (IRB) of Dongguk University Ilsan Hospital (IRB No. DUIH 2022-02-005-004), and the study was registered at the Clinical Research Information Service (CRIS, KCT0007463, date of registration: 20 June 2022). Informed consent was obtained from all participants, and all procedures were performed following the Declaration of Helsinki.

### 2.2. Participants

Thirty healthy individuals were included and randomly assigned to two groups: 15 in the CCE group (control group) and 15 in the RCE group (experimental group). The mean MMSE score was 29.47 ± 0.52 in the RCE group and 29.60 ± 0.51 in the CCE group (Table 1). The average span length of digits was 7.86 ± 0.63 in the CCE group and 7.6 ± 0.82 in the RCE group during the test. No significant differences in memory training levels were observed between the two groups (Table 1).

#### 2.2.1. Eligibility Criteria

Based on data from subjects who agreed to participate in this study, screening was conducted to select eligible subjects following the inclusion and exclusion criteria. Inclusion criteria were as follows: (1) healthy adults aged between 19 and 40 years, (2) a Mini-Mental Status Examination (MMSE-K) score of 24 or higher. Exclusion criteria were as listed: (1) individuals with cognitive or physical disabilities; (2) those with behavioral disorders that might restrain their ability to perform ‘virtual drum beating content’ and conventional memory training. Table 1 presents the baseline characteristics of the two groups.

#### 2.2.2. Sample Size

The sample size calculation was performed using data from HbO and trunk linear acceleration results obtained from the study of Mori et al. [20] (alpha = 0.05, power = 95%, effect size η^2^ = 0.34). For the current study, the estimated sample size was 15 participants for each group. We prospectively enrolled 30 suitable subjects from May 2022 to November 2022.

#### 2.2.3. Randomization Procedure

The randomization process was conducted for subjects who met the inclusion/exclusion criteria and agreed to participate in this study. They were randomly allocated to either the CCE or RCE using the computerized random number generator SAS version 9.4 (SAS Institute, Inc., Cary, NC, USA) employing the block randomization method.

### 2.3. Intervention

The CCE group underwent memory training using conventional paper-and-pencil methods in a dedicated laboratory setting. Digits were visually presented one at a time on a computer screen for five seconds. The memory span length began with three digits and gradually increased to nine. After the numbers disappeared from the screen, participants were asked to write the numbers in the correct sequential order on the response sheet [21].

The RCE group performed memory training using realistic drum beating content in an independent laboratory. Participants were asked to stand in front of a 42-inch LED monitor and memorize digits displayed on the screen for five seconds. Thereafter, they were instructed to hit the correct number of virtual drums in sequence using an electronic drumstick held in both hands. This electronic drumstick replicated the vibrations and impact of real drumming, while the monitor reproduced the drumming visual and auditory experience (Figure 1). If the participants successfully struck the correct drum for three consecutive three-digit sequences, the length of the digit span increased to nine digits. This task lasted for 60 s, and the nine-digit span continued until the end of the task. Three tasks were provided after a resting period of 30 s (Figure 2).

### 2.4. Measurement

During the intervention, we employed fNIRS to monitor changes in brain activity. The fNIRS data from NIRSIT LITE (OBELAB, Inc., Seoul, Republic of Korea) utilized near-infrared light transmission at two distinct wavelengths (780 and 850 nm). The NIRSIT obtained regulatory clearance as a medical device from the Korea Food and Drug Administration in 2017. The fNIRS results were wirelessly collected with time synchronization markers during the trial. The modified Beer–Lambert law was applied to obtain hemodynamic responses, considering the last five seconds of the pre-task as the baseline for each block [22]. We calculated the block-average responses for each channel by subtracting the mean amplitude of the baseline.

The final output of each optode was subsequently expressed as the mean total oxygenated hemoglobin (HbO μm). A total of 15 channels were positioned on both sides of the dorsolateral prefrontal cortex (DLPFC) (channels 2, 3, 12, 14), orbitofrontal cortex (OFC) (channels 1, 4, 7, 10, 13, 15), and frontopolar prefrontal cortex (FPPFL) (channels 5, 6, 8, 9, 11) [23]. These channels were situated as follows: channel 1 on the right lateral OFC; channels 4 and 7 on the right medial OFC; channels 2 and 3 on the right DLPFC; channel 15 on the left lateral OFC; channels 10 and 13 on the left medial OFC; channels 12 and 14 on the left DLPFC; and channels 5, 6, 8, 9, and 11 on the frontopolar prefrontal cortex. Figure 3 shows the positions of these channels.

The statistical analysis was performed using the NIRSIT Lite Analysis Tool v3.1.0. HbO is widely recognized for its robust and reproducible indication of changes in regional cerebral blood flow. Numerous studies have demonstrated a stronger correlation between the fMRI BOLD response and HbO as opposed to deoxyhemoglobin (deoxyHb) [24]. This is possibly attributable to the higher signal-to-noise ratio in HbO [25]. During task performance, we employed topographical maps to illustrate various regions of prefrontal cortex activity and highlight their differences. The intensity of the red color indicated an augmentation in the observed quantity of oxygenated hemoglobin within that specific prefrontal cortex area, signifying enhanced activation and cognitive engagement. On the other hand, the dark blue color represented a reduction in oxygenated hemoglobin levels and hypo-activation within the area.

### 2.5. Statistical Analysis

For demographic and clinical characteristics, categorical variables such as gender are presented as a frequency and percentage, and we assessed the pre-homogeneity through chi-square tests. Continuous variables are represented as the mean and standard deviation (SD). We analyzed subjects’ height and weight pre-homogeneity using a Student’s *t*-test, satisfying normality. For variables that did not satisfy normality, we used a Wilcoxon rank sum test. A two-sample *t*-test was used to compare group-level differences in HbO-response-accompanying events between the control and experimental groups. All values are presented as mean and SD (mean ± SD). Statistical analyses were conducted using IBM SPSS Statistics 21 (SPSS Inc., Chicago, IL, USA), and the statistical significance level was set at *p* < 0.05.

## 3. Results

All 30 participants completed the tasks without experiencing any adverse events, such as VR sickness (nausea, dizziness, or blurred vision).

Figure 4 visually presents the regions of HbO (μm) responses, highlighting increases in HbO levels (red color) and decreases (blue color). White indicates the initial state of the brain (baseline) in both the CCE and RCE groups. The statistical difference in ΔHbO (μm) of each channel is presented in Table 2. Considering all regions measured, the RCE group showed more robust HbO responses compared to the CCE group during all tasks (Figure 4). In particular, ΔHbO (μm) was statistically significant in channel 2, located on the right DLPFC, showing a mean difference of ΔHbO (μm) equal to 1.417 ± 0.616 (*p* = 0.029). In channel 7, the ΔHbO (μm) was −0.304 ± 1.234 μm for the CCE group and 0.790 ± 0.842 for the RCE group. The mean difference in ΔHbO (μm) between the CCE and RCE groups showed a significant value of 1.904 ± 0.386 (*p* = 0.008). For channel 10, the mean difference in ΔHbO (μm) was 0.592 ± 0.661 μm in the RCE group, indicating greater activation compared to the CCE group (−0.480 ± 1.229 μm). The mean difference in ΔHbO was 1.073 ± 0.360 μm (*p* = 0.006). Channels 7 and 10, which measured activations in the hypothesized medial OFC, showed significant differences between the two groups (Table 2).

## 4. Discussion

A realistic memory training exercise involving drumming was newly developed for cognitive training among older adults. We aimed to investigate prefrontal cortex activation during memory training that combines cognitive and exergame elements (dual activity). We recruited thirty healthy individuals who were randomly assigned to two groups (CCE and RCE groups). The CCE group underwent memory training using a paper-and-pencil method, a conventional approach often used in rehabilitation. In contrast, the RCE engaged in memory training by memorizing numbers displayed on a mirror-like LED screen and then matching them by beating the corresponding virtual drums. Although both interventions provided an equal level of working memory training (3–9 digits or colors), the RCE group exhibited a greater increase in brain activity compared to the CCE group.

Interestingly, our findings contrast those of Vermeij et al. [26], who observed no changes in prefrontal activation related to working memory training. Meanwhile, Belleville et al. [27], using fMRI, reported activation in frontal, temporal, and parietal areas following memory training. Additionally, they showed higher activation in the dorsolateral prefrontal cortex during memory tasks when compared to their cognitively preserved counterparts [27].

### 4.1. Different Prefrontal Cortex Activation with DLPFC

In the results, the mean ΔHbO μm in channel 2, located on the right DLPFC of the RCE group, was statistically significant when compared to the CCE group (Table 2). Activation of the DLPFC is observed when participants learn unfamiliar motor skills. The level of activation diminishes as the learning process advances [19]. Previous neuroimaging studies have provided insights into the multifaceted role of the DLPFC, which encompasses top-down attentional control [28] and dispute management [29]. However, no consensus has been found regarding the specific part within the DLPFC that acts in the above-mentioned cognitive processes. This variability in findings can be attributed to variations in control processes inherent to different task paradigms employed across prior studies [30]. Therefore, right–left differences in DLPFC activation during motor learning, including visuomotor transformation learning, may also depend on the cognitive processes involved in each motor task [28]. The DLPFC was found to be involved in the higher-order cognitive control of reaching movements, which is only needed during the early stages of visuomotor transformation learning [31]. This finding aligns with prior neuroimaging research focusing on different types of motor learning, where the author observed the activation of the DLPFC during the early stages of explicit motor learning by trial and error [32]. These studies reported differences in the right–left DLPFC activation among their participants, but the involved DLPFC side varied across studies. For instance, Sakai et al. [33] observed noticeable activation in the DLPFC starting on the left side, while continuous activation was observed in the right DLPFC compared to the left DLPFC.

On the contrary, a study examining whole-brain activity during motor sequence learning revealed the activation of the right DLPFC during the first half of learning but no activation when the task became overlearned [34]. However, it is essential to note that the significant activation of the right DLPFC observed in this study involving realistic drum content does not necessarily imply that the activation was in the learning stage. Instead, this heightened activation may be indicative of a compensatory mechanism enabling individuals to perform adequately, particularly in cases where memory networks may not be functioning optimally [35]. The DLPFC is implicated in recognizing salient events, both stimulating and inhibiting motor responses, shifting attentional focus, and directing attention to new targets [36]. Moreover, it plays a critical role in monitoring stimulus occurrence, leading to a decrease in reaction time as the inter-stimulus interval (ISI) increases (for period effect), a function also reliant on the integrity of the DLPFC [37]. Nonetheless, recent evidence suggests that the right DLPFC may operate more efficiently. It sustains attention and maintains a heightened state of alertness, induced by warning signals that briefly precede the target. These processes may be associated with the anterior cingulate cortex (ACC) or other midline frontal structures [38].

### 4.2. Different Prefrontal Cortex Activation with OFC

The hypothesized medial OFC (channels 7, 10) within the prefrontal lobe showed more activations in the RCE group during memory training with the virtual drum. Previous NIRS studies have demonstrated that the prefrontal cortex, including the OFC, becomes activated during tasks requiring verbal fluency [39,40]. The OFC has been previously associated with reward processing [41]. Touch, as a primary reinforcer, can evoke feelings of pleasure [42]. Notably, fMRI studies conducted by Rolls and colleagues [43] have demonstrated that reward-related brain regions, including the OFC, are activated in response to pleasant touch sensations, such as those by velvet fabric, which aligns with our present results. Furthermore, another research study on fMRI has also reported the activation of similar brain regions in response to pleasant touch experiences [44]. A controller (HTC Vive Pro, HTC, Taiwan) without an HMD was used as a display device for the execution of VR drum content. These controllers were equipped with vibration motors designed to provide tactile feedback in the form of vibrations. Research has demonstrated that incorporating vibration as a source of force feedback can yield improvements compared to relying solely on visual feedback [45]. The human sense of touch is particularly adept at detecting and interpreting these vibrations, which provide valuable information about interactions with the physical world [46]. Tactile sensations such as pressure and vibration are perceived by specialized sensory receptors called mechanoreceptors implanted into the skin. Each type of mechanoreceptor is specialized to perceive and respond to a specific type of tactile stimuli [47,48].

### 4.3. Strengths and Limitations

Our study has several strengths. These include a simple design with a short intervention period, a randomized trial design, and the distinction of being the first clinical study of the newly developed virtual drum beating system. In addition, this study investigated the effectiveness of the prefrontal cortex in response to the unique characteristics and requirements of older adults. However, it is crucial to acknowledge the limitations of this preliminary study. While the content and equipment of virtual drum beating represent a cognitive training method for older adults, the outcomes cannot be generalized to the elderly population due to the preliminary nature of this research conducted on healthy people (healthy adults aged 19 to 40 years). Before starting the study, we did not identify any prior studies using the same intervention (realistic cognitive exergame with virtual drum beating content) with the same primary outcome of HbO. Consequently, the sample size of 30 participants may have been insufficient for robust conclusions. Further studies should aim to include a more representative and larger sample size, specifically targeting older adults, building upon the insights gained from this initial investigation.

## 5. Conclusions

Recently, a novel virtual drum beating system was developed, capable of simultaneously training memory skills while engaging in drumming activities. This study used functional near-infrared spectroscopy (fNIRS) to examine the effects of realistic cognitive and behavioral tasks on brain activity. The primary outcome measure was the level of oxyhemoglobin in the dorsolateral prefrontal cortex (DLPFC). In the right DLPFC, a statistically significant mean difference in HbO of 0.417 ± 0.616 μm (*p* = 0.029) was observed. Notably, remarkable activations were recorded in the medial OFC on channels 7 and 10. The combination of memory training and virtual drumming provided distinct sensory inputs, encompassing visual, auditory, and vibrational cues. This stimulation resulted in heightened activity in the DLPFC and OFC regions for the RCE group. When compared to conventional cognitive training, the new memory training program, incorporating realistic cognitive exergaming with a virtual drum component, demonstrated a greater degree of brain activation, particularly in oxyhemoglobin levels within the VLPFC and OFC.

## Figures and Tables

**Figure 1 healthcare-11-02559-f001:**
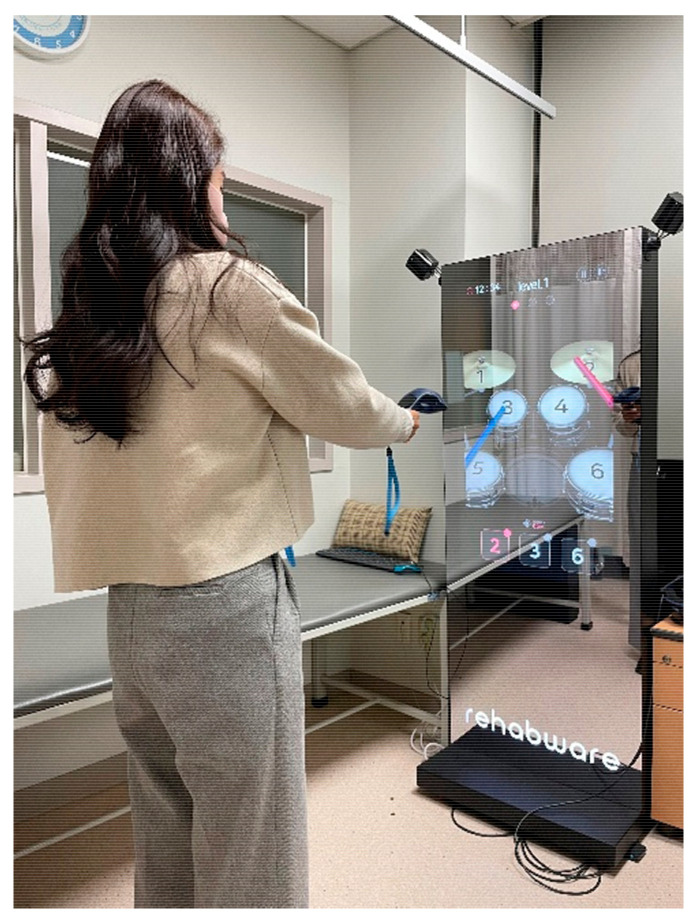
Memory training by virtual drum beating.

**Figure 2 healthcare-11-02559-f002:**
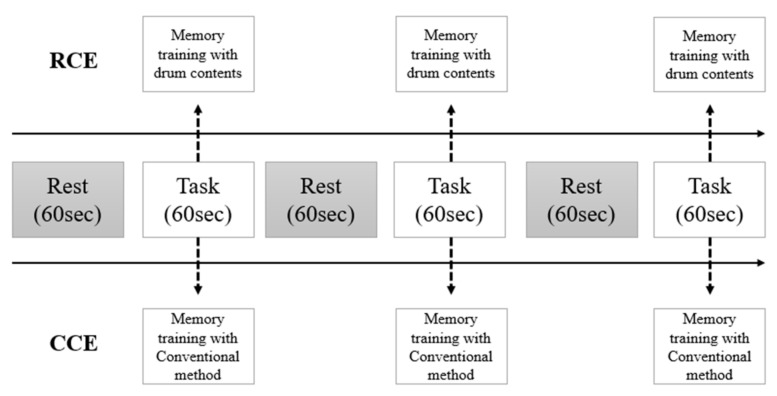
The protocol of measurement with fNIRS.

**Figure 3 healthcare-11-02559-f003:**
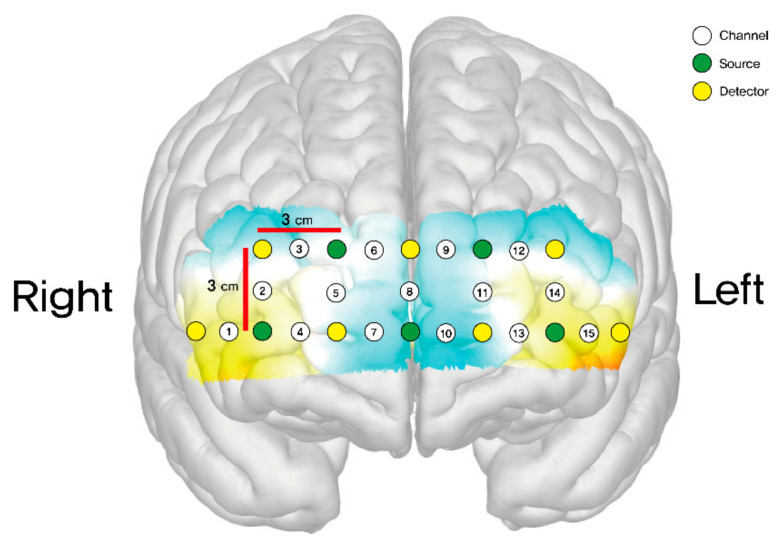
Schematic positions of fNIRS channels.

**Figure 4 healthcare-11-02559-f004:**
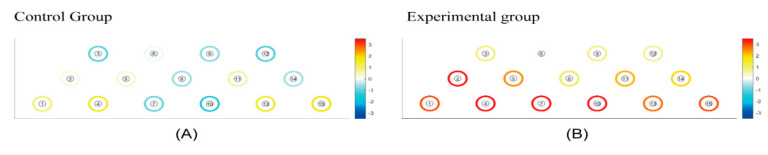
(**A**) The mean hemodynamic changes (ΔHbO) when performing three instances of conventional cognition training. (**B**) The mean hemodynamic changes (ΔHbO) when taking three periods of rest between pre and post experimental cognition training. White: initial state of the brain (baseline). White < yellow < red: increase in HbO level. White > light Blue > blue: decrease in HbO level.

**Table 1 healthcare-11-02559-t001:** Baseline characteristic of subjects in the CCE and the RCE group.

Variables	CCE *n* = 15	RCE *n* = 15	*p*-Value
Sex, *n* (%)			
Male	7 (58.3%)	5 (41.7%)	0.456 *
Female	8 (44.4%)	10 (55.6%)	
Age (Mean ± SD)	29.47 ± 3.56	29.07 ± 4.13	0.779 ^#^
Height (cm)	167.20 ± 8.12	167.07 ± 8.18	0.965 ^#^
Weight (kg)	64.60 ± 18.05	58.07 ± 13.26	0.269 ^#^
MMSE	29.60 ± 0.51	29.47 ± 0.52	0.481 ^##^
Number of digits	7.86 ± 0.63	7.60 ± 0.82	0.401 ^#^

CCE—conventional cognitive exercise group, RCE—realistic cognitive exergame group, MMSE—Mini-Mental State Examination, number of digits—peak performance of memory training during intervention. *: *p*-value obtained from chi-square test; ^#^: *p*-value obtained from Student’s *t*-test; ^##^: *p*-value obtained from Wilcoxon rank sum test.

**Table 2 healthcare-11-02559-t002:** Differences in HbO when undergoing cognitive memory training in both groups (μm).

Gus	CCE	RCE	Difference	*p*
	*n* = 15	*n* = 15		
1	0.370 ± 1.532	1.202 ± 1.555	0.832 ± 0.564	0.151
2	0.104 ± 1.771	1.521 ± 1.598	1.417 ± 0.616	0.029 *
3	−0.344 ± 1.291	0.228 ± 0.944	0.572 ± 0.413	0.177
4	0.470 ± 1.142	0.963 ± 0.909	0.492 ± 0.377	0.202
5	0.139 ± 1.695	0.533 ± 0.815	0.394 ± 0.486	0.424
6	−0.032 ± 1.494	−0.002 ± 0.787	0.030 ± 0.436	0.946
7	−0.304 ± 1.234	0.790 ± 0.842	1.094 ± 0.386	0.008 *
8	−0.295 ± 1.813	0.284 ± 0.922	0.579 ± 0.525	0.28
9	−0.277 ± 1.436	0.260 ± 1.210	0.537 ± 0.485	0.278
10	−0.480 ± 1.229	0.592 ± 0.661	1.073 ± 0.360	0.006 *
11	0.248 ± 1.491	0.540 ± 0.932	0.293 ± 0.454	0.524
12	−0.419 ± 1.405	0.216 ± 1.150	0.635 ± 0.469	0.186
13	0.423 ± 1.055	1.228 ± 1.737	0.805 ± 0.525	0.136
14	−0.148 ± 1.011	1.105 ± 2.166	1.253 ± 0.617	0.052
15	0.679 ± 1.399	2.284 ± 2.953	1.605 ± 0.844	0.067

CCE—conventional cognitive exercise group, RCE—realistic cognitive exergame group, * *p* < 0.05 by two-sample *t*-test.

## Data Availability

The data presented in this study are available on reasonable request from the corresponding author.

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
