# Peer review of "Prefrontal Cortex Activation during Memory Training by Virtual Drum Beating: A Randomized Controlled Trial"

_healthcare, 2023, doi:10.3390/healthcare11182559_

Round 1

Reviewer 1 Report

This study aims to investigate the differences in brain activities during memory training using conventional methods and a realistic cognitive exergame that does not cause sickness. However, there are still a few issues that need to be addressed.

 Firstly, the Introduction lacks sufficient detail and fails to present necessary research theories, reasonable hypotheses, or research purposes. For instance, although previous articles have demonstrated positive effects of VR game interventions on attention, visual and verbal memory, and executive functioning in older adults with MCI (line 42), these studies were not included in the paper, and the potential side effects of VR disease were not explained. Furthermore, the main outcome measure, oxyhemoglobin level in DLPFC, and the activations in the hypothesized medial OFC were not mentioned in the introduction.

 Secondly, the Materials and Methods section does not provide clear explanations. It is recommended to separate the introduction of participants, procedures, and evaluation criteria to enhance readability. Additionally, detailed information about the participants, such as the sex ratio and average age, was not provided, and a sample size of 15 subjects in each group appears to be small.

Thirdly, in the Results section (Section 3), the significance of p-values should be mentioned. Furthermore, the symbols "*/#/##" used in Table 1 to represent different detection methods are not explained in the table.

Finally, it is suggested that the explanatory information in the Discussion (e.g., the introduction to fMRI and fNIRS, lines 181-194) be moved to the beginning of the Introduction for better clarity."

Author Response

We appreciate your review which allowed us to greatly improve the quality of the manuscript. We agree with all your comments, and we corrected point by point the manuscript accordingly and underwent intensive English editing. We would be grateful if the revised manuscript could be considered for publication in Healthcare, MDPI.

Sincerely

YG Nam and BS Kwon.

*Description line number is the number of the final (changes are applied) version of the edit.

Reviewer 2 Report

VR material is being used more and more for neurological illnesses accompanied by cognitive impairment. We created a virtual drum-beating system that could concurrently train the memory of numbers and beat the drum. This study used functional near-infrared spectroscopy (fNIRS) to examine the effects of realistic cognitive and behavioral tasks on brain activity. The two groups of traditional cognitive exercise (CCE) and realistic cognitive exercise (RCE) were randomly allocated to 30 healthy individuals. The CCE group practiced their memory by memorizing the number displayed on the computer screen and writing it down. Oxyhemoglobin level in the dorsolateral prefrontal cortex (DLPFC) served as the primary outcome indicator. The average number of digits in the results was 7.86 0.63 in CCE and 7.6 0.82 in RCE. The channel 2 on the right DLPFC had a mean differential of HbO of 0.00142 0.00062 m (p = 0.029). Also excellent were the measurements of activations in the putative medial orbitofrontal cortex (OFC) on channels 7 and 10. Because simultaneous memory training and virtual drumming supplied different sensory inputs, including visual, auditory, and vibration of the controller, the RCE group's DLPFC and OFC were more stimulated.

1. It is not clear, why a small number of people was used to conduct such an analysis. Any previous reference that can be used for such numbers?

2. how the authors have calculated the number of subjects for the experimental groups?

3. The research gap is not visible in the Introduction section. It is necessary to provide a justification for why this work is needed. 

4. p-values in the Table showed what? What is its importance herein?

5. Whether the fMRI bold response is correlated with HbO or HbR? Please provide the reference. 

6. Which statistical test was applied? It is not evident from the Materials section.

7. The obtained results are not compared with the existing results. The authenticity of the wok needs to be validated with such comparisons. 

8. There is no conclusion section. It is difficult to see the output. 

9. The HbO results are not visible in the figure. The figure that contains such a curve along with STD must be there with an explanation. The authors are encouraged to see the existing research within the country and/or outside.

10. The references are poorly managed that need corrections. Please double check. 

Author Response

(The authors gave the same response as above.)

Reviewer 3 Report

Review Results

Manuscript entitled: Prefrontal cortex activation during memory training by virtual 2 drum beating: A randomized controlled trial

In this manuscript, the authors stated that virtual reality (VR) content is increasingly used in neurological disorders with cognitive impairment. So, they developed a device of virtual drum beating content that could provide memory of digits training with drum beating simultaneously in order to investigate the effects of realistic cognitive and brain activity by functional near-infrared spectroscopy (fNIRS). They found that DLPFC and OFC were more activated in the RCE group (p < 0.05) because simultaneous memory training and virtual drum beating provided various sensory inputs of visual, sound, and vibration from the controller.

Lines 69–70: The authors stated that... "Randomization tables were created for each research organization." What does this statement mean?

Lines 84–89: The authors stated that... "Based on the data of patients who agreed to participate in this study, screening was conducted to select eligible subjects based on the following inclusion and exclusion criteria: Inclusion criteria were: 1) healthy adults aged 19 to 40 years; and 2) a Mini-Mental Status Examination (MMSE-K) score of 24 or higher. Exclusion criteria were: 1) those with cognitive or physical disabilities; 2) those with behavioral disorders that might make it difficult to perform virtual drum beating content' and conventional memory training."

I do not understand why the authors used different terms for subjects who participated in this study. Did they study with patients or healthy adults?

Table 1: I could not find any symbols representing the following indicators:  

*: p-value obtained from Chi-square test;

#: p-value obtained from Student's t-test;

##: p-value obtained from Wilcoxon rank sum test.

Figure 3: Is it possible to describe the statistical analysis in Figure 3?

Lines 172-173: The authors stated several times in several parts of the manuscript that "A realistic memory training content of drum beating was newly developed for cognitive training for the elderly."

However, participants aged 17–40 were selected in both groups. How do these participants represent the significant outcome of the realistic memory training contents of drum beating for the elderly, as this study did not apply to the elderly at all?

There is no conclusion in both the abstract and at the end of this manuscript. The explanation of the mean difference of ΔHbO in channel 2, located on the right DLPFC, and those in channel 7 and channel 10, hypothesized medial orbitofrontal cortex (OFC), is needed to be explained in more detail, especially in the selected participants and the elderly.   

Finally, as the percentage of similarity is too high (52%), it is not appropriate for the current version of the manuscript to be considered for publication. 

Author Response

(The authors gave the same response as above.)

Reviewer 4 Report

Thank you for your submission. 

This research reports the outcomes of memorized digits and oxygenation via fNIRS in response to both conventional cognitive exercise and realistic cognitive exergame. The reviewer does feel that there are major changes needed prior to publication. In addition, the reviewer does recommend that a native English speaker revises the entire manuscript prior to resubmission as there are many mistakes throughout from an English language standpoint (e.g., it seems some sentences don't end - line 127 There¬fore, only HbO derived concentration changes [18].).

Introduction - you speak a lot about the elderly & MCI in the opening paragraphs, but your study was conducted in healthy adults aged 19-40 years. Therefore, I think you need to rethink the structure to your introduction for a more simplistic and straightforward read for your audience. 

Methods - so was it only one trial during which fNIRS was measured? It's difficult to understand the learning so perhaps a figure would help the reader visualize the specifics of the study. 

Is the caption in figure 3, correct? Figure B also says pre- and post- conventional cognition training?

Results - the changes in oxyhemoglobin are extremely small (out to three or four decimal places).

Discussion - check reference line 198

If the RCE group produced greater activation of the right DLPFC (channel 2) (similar for channels 7 & 10), doesn't this mean that this area of the brain was working harder during the exergame trial? If so, what does this mean from a learning standpoint?

You talk about stages of learning but wasn't this an acute study?

Limitations and separate conclusion paragraph?

Needs significant work as there are many grammatical errors throughout - line 56: The controller used to perform contents.

Author Response

(The authors gave the same response as above.)

Round 2

Reviewer 1 Report

The authors have made significant changes and improvements and have addressed all of my previous comments and suggestions. I believe that the quality of the manuscript has been enhanced.

Author Response

Response to Reviewer #R1

We appreciate your decision. We checked again every word and grammar errors in the manuscript. We would be grateful if the revised manuscript could be considered for publication in Healthcare, MDPI.

Sincerely,

YG Nam and BS Kwon.

* All changes are highlighted in yellow.

Reviewer 2 Report

Thr authors have addressed my comments well.

Author Response

We appreciate your decision. We checked again every word and grammar errors in the manuscript. We would be grateful if the revised manuscript could be considered for publication in Healthcare, MDPI.

Sincerely,

YG Nam and BS Kwon.

* All changes are highlighted in yellow.

Reviewer 3 Report

Thank you for clarifying all of my concerns.

Good luck.

Author Response

(The authors gave the same response as above.)

Reviewer 4 Report

Thank you for your resubmission. 

Firstly, I think the manuscript would have better suitability in other journals focused on the brain or psychology. That said, I think there are also still errors that need editing prior to resubmission anywhere. I've never seen such a hemodynamic response as small as the numbers you are reporting for the changes in activation? In your abstract these numbers are out to five decimal places and in the table only to four. There are also still some English grammar errors. I'm also not sure why you didn't include limitations and a conclusion the first time around. In table 2 there are several places where there is a significant difference and there is no * or indication of these differences at all. I think you need to take your time and go through every word prior to resubmission (e.g., HBO in some places and HbO in others).

Better but not up to publication standard yet.

Author Response

We appreciate your review. We agree with all your comments and our research team did our best to r double-check for whole manuscript. We would be grateful if the revised manuscript could be considered for publication in Healthcare, MDPI.

Sincerely

YG Nam and BS Kwon.

All changes are highlighted in yellow.

We rechecked every word and grammar errors in the manuscript. We checked the results part of hemodynamic response and the figures currently written in “mm(mMol)” were converted into “μm(uMol)” units and presented in whole manuscript and Table 2 from the recent reference*. And we check the indication for significant difference in Table 2. We will remember your review in the next study, really appreciate again your comments.

*Yoon, Jin A., et al. "Neural compensatory response during complex cognitive function tasks in mild cognitive impairment: a near-infrared spectroscopy study." Neural Plasticity, 2019
